# Application of Continuous Wavelet Transform and Artificial Naural Network for Automatic Radar Signal Recognition

**DOI:** 10.3390/s22197434

**Published:** 2022-09-30

**Authors:** Marta Walenczykowska, Adam Kawalec

**Affiliations:** Faculty of Mechatronics, Armament and Aerospace, Military University of Technology, 00-908 Warsaw, Poland

**Keywords:** radar signal recognition, artificial neural network (ANN), continuous wavelet transform (CWT), automatic signal recognition (AMR), feature extraction

## Abstract

This article aims to propose an algorithm for the automatic recognition of selected radar signals. The algorithm can find application in areas such as Electronic Warfare (EW), where automatic recognition of the type of intra-pulse modulation or the type of emitter operation mode can aid the decision-making process. The simulations carried out included the analysis of the classification possibilities of linear frequency modulated pulsed waveform (LFMPW), stepped frequency modulated pulsed waveform (SFMPW), phase coded pulsed waveform (PCPW), rectangular pulsed waveforms (RPW), frequency modulated continuous wave (FMCW), continuous wave (CW), Stepped Frequency Continuous Wave SFCW) and Phase Coded Continuous Waveform (PCCW). The algorithm proposed in this paper is based on the use of continuous wavelet transform (CWT) coefficients and higher-order statistics (HOS) in the feature determination of selected signals. The Principal Component Analysis (PCA) method was used for dimensionality reduction. An artificial neural network was then used as a classifier. Simulation studies took into account the presence of noise interference with signal-to-noise ratio (SNR) in the range from −5 to 10 dB. Finally, the obtained classification efficiency is presented in the form of a confusion matrix. The simulation results show a high recognition test accuracy, above 99% with a signal-to-noise ratio greater than 0 dB. The article also deals with the selection of the type and parameters of the wavelet. The authors also point to the problems encountered during the research and examples of how to solve them.

## 1. Introduction

The modern battlefield requires both an effective threat detection system and a system enabling their correct classification. The richness of signals in the radio space means that, both in the case of ELINT and ES, which are elements of EW [1], the development of an effective and fast algorithm for recognizing enemy radars is a key element contributing to the success of the mission. Fast and appropriate recognition of the signal type and the related ability to identify the sources of emissions and the mode of their work is one of the key aspects determining the deployment of forces in military operations. When it comes to military navigation, the efficient ES allows to recognise and avoid dangers in a timely manner. The systems, which provide knowledge of the environment and the deployment of military forces and assets, allow a safe route to be determined, which is particularly important when navigating in areas of military operations. It can also help to identify navigation systems interferences. That is why radio-electronic reconnaissance is an integral part of the fire command and control systems.

In electronic intelligence (ELINT) systems, many emission source parameters, such as radio frequency (RF), direction of arrival (DOA), time of arrival (TOA), pulse width (PW), pulse repetition frequency (PRF), intra-pulse modulation, etc. are determined. In paper [2] an instantaneous frequency profile is used to measure the exotic modulations and their parameters. The paper considers linear frequency modulation (LFM), bi-phase modulation (BPM) and stepped frequency modulation (SFM).

The problem of measuring the parameters of specific radar signals in real time still remains a challenge. The emission identification process is most often carried out using a knowledge-based approach. Paper [3] discusses methods for determining specific radar signal parameters (signatures).

A sorting algorithm for pulse radar based on wavelet transform is proposed in [4]. The authors of the study point to the key role of the accuracy of TOA determination, the estimation of which affects the correctness of the determination of parameters such as PW and PRF. They propose the use of the wavelet transform and Haar wavelet as a tool for sorting radar signals.

A method of recognizing four types of radar signals with three characteristic parameters determined from: CWT eigenvalue, frequency domain moment, kurtosis coefficient, and frequency domain moment skewness coefficient are presented in [5]. The following signals were considered by authors: polynomial phase signal, pseudo code phase modulation and sinusoidal frequency modulation, product composite of pseudo code phase modulation and LFM, convolution composite of pseudo code phase modulation and LFM. For signal to noise ratio (SNR) higher than 0 dB, the probability of signal recognition is greater than 98%.

A proposal for a radar signal recognition method, this time based on the time-frequency transform and high order spectra analysis, is presented in [6]. Time-frequency image (TFI) determined using Wigner–Ville Distribution (WVD) is discussed in [7].

The interference classification and recognition based algorithm is presented in [8] and considers recognition of single-tone interference, multi-tone interference, narrow-band interference and the linear chirp, and impulse and comb spectrum interference.

Another aspect important from the point of view of the real-time recognition process is the classification method. Currently, due to the dynamic development of the field of machine learning, it is more and more often proposed to use artificial neural networks (ANN) as a classifier [9,10]. Using feedforward networks is proposed in [11] and convolutional neural networks (CNN) in [12], usually with features in the form of an TFI, are discussed in [7,13,14,15].

The signal recognition process typically involves pre-processing the radar signal and its analysis to identify signal features, that allow the emitter to be identified. Due to the increasing number of new types of radar signals and their complexity, the electromagnetic environment is becoming more difficult to analyse. Hence, the need to research and develop new algorithms with a wide spectrum of recognition, becomes a pressing problem to solve.

The aim of the article is to present an algorithm for automatic recognition of selected radar signals. The proposed solution based on the use of CWT and HOS is characterised by both high flexibility (related to the variety of possible wavelets and the selection of their parameters) and versatility. A significant advantage of the analysed radar signal classification algorithm is also the possibility of using CWT at individual stages of signal analysis, e.g., to estimate selected parameters of a radar signal. These conditions are met by the feature extraction methods using CWT, which is successfully used in algorithms for sorting radar pulses and determining parameters such as pulse repetition time (PRT), pulse width (PW) or time of arrival (TOA) [5].

A very important aspect discussed in the article is also the influence of parameters used in simulations of analytical wavelets on the properties of the proposed algorithm. Thanks to the analysis of the aforementioned parameters, the algorithm achieves additional adaptation possibilities depending on the specificity of the signals that will be taken into account during the classification. The usefulness of CWT is mainly due to the ability to observe the instantaneous amplitude, phase and frequency of signals simultaneously, without having to plot each of these parameters separately in the time domain. The appropriate selection of the wavelet allows, in turn, to obtain the required resolution in time and frequency, and to emphasise certain features of the analysed signals.

At the same time, the necessity to make online decisions requires the use of classifiers that meet specific time requirements. Therefore, it seems obvious to use artificial intelligence techniques. Although the process of training such a model is time-consuming and requires a large number of training vectors, the trained model allows for signal classification in immediate mode.

The article presents the process of recognizing selected types of radar signals using CWT, HOS and ANN, taking into account analytical wavelets and the influence of their parameters on the obtained results. This new approach to the process of recognizing radar signals allows for the flexibility of the algorithm and the possibility of its comprehensive application by changing parameters such as the signal observation time or the type and parameters of the wavelet used.

So far, the simulations necessary to test the proposed algorithm and confirm its effectiveness in the Matlab environment have been carried out. Ultimately, however, the authors plan to test the algorithm on real signals.

## 2. Materials and Methods

The effectiveness of algorithms using CWT and ANN is confirmed by research into algorithms for recognising modulations typical of signals used in communications such as amplitude shift keying (ASK), phase shift keying (PSK), frequency shift keying (FSK) or quadrature amplitude modulation (QAM) [16,17,18].

The algorithm proposed in this work identifies specific radar signals (Figure 1): Linear Frequency Modulated Pulsed Waveform (LFMPW), Stepped Frequency Modulated Pulsed Waveform (SFMCW), Phase coded pulsed waveform (PCPW), Rectangular pulsed waveform (RPW), Frequency Modulated Continuous Wave (FMCW), CW (Continuous Wave), SFCW (Stepped Frequency Continuous Wave) and Phase Coded Continuous Waveform (PCCW).

This section will present the methodology for determining radar signals features using CWT, HOS and ANN and the overall structure of proposed radar signal recognition process.

### 2.1. The Continuous Wavelet Transform (CWT) of Radar Waveforms

The continuous wavelet transform (CWT) enables extraction of transient information associated with amplitude/frequency changes and phase shifts which are characteristics of modulated signals. CWT of a signal *x(t)* is defined as [19]:(1)$$CWT\left(\tau ,a\right)=\int x\left(t\right)\psi_{a}^{*}\left(t\right)dt=\frac{1}{\sqrt{\left|a\right|}}\int x\left(t\right)\psi^{*}\left(\frac{t-\tau }{a}\right)dt$$
where the ψ(t) is called mother wavelet and *a* is the scaling constant. $\psi_{a}^{*}$ is called baby wavelet and is the translated and scaled version of ψ(t).

In case of digital implementation of CWT, the integral in Equation (Equation 1) can be replaced by summation. Assuming that *t = kT = k, z = nT = n* and scale is an even integer, we could write that [18]:(2)$$CWT\left(n,a\right)=\frac{1}{\sqrt{a}}\sum x\left(k\right)\psi^{*}\left(\frac{k-n}{a}\right)$$

The simulation studies analysed the use of the following analytical wavelets: generalised Morse, Bump and Morlet. The properties of the Bump wavelet are described in [20]. The generalised Morse wavelet, according to [20,21,22], is defined as follows:(3)$$\psi_{\beta ,\gamma }\left(\omega \right)=U\left(\omega \right)a_{\beta ,\gamma }\omega^{\beta }e^{-\omega^{\gamma }}$$
where $a_{\beta ,\gamma }$ is a normalization constant, U(ω) is the unit step function, and β and γ are parameters controlling the wavelet form. Normalization constant is defined [21]:(4)$$a_{\beta ,\gamma }\equiv 2{\left(\frac{e\gamma }{\beta }\right)}^{\frac{\beta }{\gamma }}$$

It follows from the analysis presented in the [20] that by varying β and γ parameters, the generalised Morse wavelets can take a wide variety of forms. For example the γ=1 family corresponds to the Cauchy (or Paul) wavelet, the γ=2 correspond to analytic Derivative of Gaussian wavelets, and γ=3 corresponds to the Airy wavelets family [21].

In Matlab Wavelet Toolbox parameters for a parameterised analytic Morse wavelet are defined as γ, the time-bandwidth product $P^{2}=\beta \gamma$ and they correspond to the analysis in [20,21,22].

Based on the analysis presented above, simulation studies were carried out to select a suitable wavelet to best reflect the characteristics of the recognised signals. We use the scalogram, which is the absolute value of the CWT of a signal, plotted as a function of time and frequency. Scale is assumed to be proportional to the inverse of the frequency [21].

Figure 2, Figure 3 and Figure 4 show the scalogram for LFMCW, PCCW and SFMCW using the following waveforms:Morse with parameters γ=27, β=27 presented in [20];Bump;Morse with default parameters γ=3, β=20;

The best resolution in frequency was observed for the Morse wavelet with parameters γ=27, β=27. The use of a Morse wavelet with the default parameters results in a scalogram which is fuzzy in frequency. The scalograms for PCCW clearly show the moments at which the phase shift occurred. In the case of the Morse wavelet with the default parameters, the vertical lines corresponding to phase shifts are clearly visible over the entire frequency range.

It would seem, therefore, that the use of the Morse wavelet with default parameters values is appropriate for the identification of phase-coded signals. However, it is clear from Figure 5 that, in the case of stronger noise interference, the use of this wavelet will no longer be effective. For the Bump and Morse with γ=27, β=27 moments of phase change are still visible.

For SFM waveforms the Morse wavelet with γ=27, β=27 is recommended for best frequency resolution (Figure 5).

In order to reflect the characteristics of the individual types of waveforms and to enable their identification after calculating the CWT coefficients, frequency cuts for selected scales can be used, as shown in [17]. Using a median filter for smoothing purposes gives relevant features for classification [23]. The frequency cuts for frequency close to signal carrier frequency are presented in Figure 6.

In the case of signal recognition algorithms, where the CWT is used for feature extraction, an important parameter influencing the correctness of the classification is the scale a. The problem of how to choose the optimal scale for modulation recognition purposes has been addressed in papers [23,24,25]. One of the least complicated solutions to the scale selection problem consisting in summing CWT coefficients for several selected scales (frequencies) was proposed in [24].

During the simulation studies, it was assumed that the exact value of the carrier frequency and bandwidth for a given signal were unknown and needed to be estimated. In order to determine the characteristics of the signals, the CWT frequency cut $\left|CWT(f_{0},t)\right|$ and the sum of the CWT coefficients for the two selected frequency (scale) ranges: $\Delta f_{1}$ and $\Delta f_{2}$ are used. Whereby the $f_{o}≅f_{c}$ and $f_{c}$ is estimated signal carrier frequency. Examples of CWT frequency cut and sum of CWT coefficients for PCPW, SFCW nad LFMCW are presented respectively in Figure 7, Figure 8 and Figure 9.

### 2.2. Higher Order Statistics as Signal Features

Higher order statistics (HOS) for feature extraction purposes are considered in [17,26,27]. To determine the characteristics of the specific signals, selected statistical parameters, such as mean value, standard deviation, moments and cumulants of higher orders are proposed. These parameters calculated for CWT frequency cut and summed CWT coefficients for $\Delta f_{1}$ and $\Delta f_{2}$ are part of features on the basis of which the classification process is carried out in the proposed recognition algorithm. The cumulants of any order *n*, assuming that the real signal is analysed, can be calculated using the following expression [27]:(5)$$C_{n}=M_{n}-\sum_{m=1}^{n-1}\frac{(n-1)!}{(m-1)!(n-m)!}C_{m}M_{n-m}$$
where the moment of *n* is defined as follows:(6)$$M_{n}=\frac{1}{N}\sum_{k=1}^{N}{\left(x-\mu \right)}^{n}$$
where μ is the mean value of the random variable *x*. According to the [27] for Gaussian zero-mean random variables, odd numbered cumulants are zero.

In turn, the following expressions for determining higher order mixed moments for complex variables are presented in [26]:(7)$$M_{pq}=E[\left(x{\left(n\right)}^{p-q}x^{*}{\left(n\right)}^{q}\right]$$
where * denotes conjugate, *q* is an integer less than *p/2*, *p* is an order of moment.

According to [28] cumulant of n variables $x_{1},\ldots ,x_{n}$ is defined as:(8)$$Cum\left(x_{1},\ldots ,x_{n}\right)=\sum_{\pi }\left(|\pi |-1\right)!{\left(-1\right)}^{|\pi |-1}\prod_{B\in \pi }E\left(\prod_{i\in B}x_{i}\right)$$
where the |π| goes through the list of all partitions $\left\{x_{1},\cdots ,x_{n}\right\}$, *B* goes through the list of all blocks of the partition |π|, E( .)—expectation.

Second, fourth and sixth order cumulants can be represented as [26,29]:(9)$$C_{20}=Cum\left(x,x\right)=M_{20}$$
(10)$$C_{21}=Cum\left(x,x^{*}\right)=M_{21}$$
(11)$$C_{40}=Cum\left(x,x,x,x\right)=M_{40}-3M_{20}^{2}$$
(12)$$C_{41}=Cum\left(x,x,x,x^{*}\right)=M_{41}-3M_{21}M_{20}$$
(13)$$C_{42}=Cum\left(x,x,x^{*},x^{*}\right)=M_{42}-{\left|M_{20}\right|}^{2}-2M_{21}^{2}$$
(14)$$C_{63}=M_{63}-9C_{42}C_{21}-6{\left(C_{21}\right)}^{3}$$

According to [28] cumulants possess one ability which is not present in signal moments: if a variable follows Gaussian statistics, all its cumulants of order higher than two are equal to zero. That is why they are quite often used in automatic modulation recognition context, since they are directly applicable to Gaussian noise. In [28,30] normalised higher-order cumulants are proposed and this form of cumulants was used in our simulation studies.

When calculating the features of modulated signals with the use of cumulants, it is worth using the concept of normalised cumulants mentioned in the article [28,30]:(15)$$feature_{1}=\frac{|C_{40}|}{|C_{21}{|}^{2}}$$
(16)$$feature_{2}=\frac{|C_{41}|}{|C_{21}{|}^{2}}$$
(17)$$feature_{3}=\frac{|C_{42}|}{|C_{21}{|}^{2}}$$
where $C_{21}$ is the signal energy.

### 2.3. Algorithm Principle

The proposed algorithm for recognising selected radar signals is shown in Figure 10 and consists of the following steps:1.signal pre-processing;2.signal features extraction based on CWT coefficients, HOS and additional signal features form CWT frequency profile;3.feature normalization and reduction of dimensions with PCA;4.classification using ANN.

The process of determining the features is as follows:calculation of the CWT coefficients for the frequency range $\Delta f_{all}$;determination of CWT frequency cut for $f_{0}≅{\tilde{F}}_{c}$ and its modulus;calculating the sum of the CWT coefficients for each frequency range:$\sum_{\Delta f_{1}}CWT\left(f,\tau \right)$, $\sum_{\Delta f_{2}}CWT\left(f,\tau \right)$, $\sum_{\Delta f_{all}}CWT\left(f,\tau \right)$and the sum of their moduli:$\sum_{\Delta f_{1}}\left|CWT\left(f,\tau \right)\right|$, $\sum_{\Delta f_{2}}\left|CWT\left(f,\tau \right)\right|$, $\sum_{\Delta f_{all}}\left|CWT\left(f,\tau \right)\right|$;calculating the mean values and standard deviations for the:$\sum_{\Delta f_{1}}\left|CWT\left(f,\tau \right)\right|$, $\sum_{\Delta f_{2}}\left|CWT\left(f,\tau \right)\right|$, $\sum_{\Delta f_{all}}\left|CWT\left(f,\tau \right)\right|$both with and without the median filter applied;determination of joint moments and cumulants for the:$\sum_{\Delta f_{1}}CWT\left(f,\tau \right)$, $\sum_{\Delta f_{2}}CWT\left(f,\tau \right)$, $\sum_{\Delta f_{all}}CWT\left(f,\tau \right)$;additional signal feature form CWT frequency profile defined as: $\sum_{\tau }CWT\left(f,\tau \right)$.

Since CWT coefficients are complex in the calculation of higher order statistics, the Equations used were (Equation 6), (Equation 8)–(Equation 14). Formulas for real signals (Equation 4) and (Equation 5) can also be used, assuming that they are calculated for CWT coefficient modules. However, this option was not considered in the current study.

The number of determined features is 48 statistical parameters and one additional feature related to the number of profile maxima, for a total of 49. The use of the CWT frequency profile and the calculation of the number of maxima is due to the need for additional features to isolate SFM signals from LFM signals. The suitability of the CWT frequency profile is closely linked to the CWT frequency resolution and therefore to the choice of wavelet type and its parameters. If other types of wavelets are to be used, it is necessary to analyse whether the wavelet provides sufficient frequency resolution first.

The idea of using a profile is used in the analysis of images and is presented, among others, in the work of [31]. This is used for the methods proposed to protect documents against copying. By calculating a profile of a document page in the form of an image, it is possible to determine the individual lines and then to determine the position of the individual words for a given line. According to [31] a profile is the projection of a two-dimensional array into a single dimension. Example of frequency profiles for LFMCW, SFCW and PCCW respectively, are shown on Figure 11, Figure 12 and Figure 13. By analysing the shape of the frequency profile and finding the local maxima of greatest prominence, number of frequencies in multi-carrier signals, like SFM, can be established. The prominence of a local maxima reflects how the peak stands out with respect to its height and location relative to other peaks.

Once all the features have been calculated, normalisation is carried out and then dimensionality reduction is performed using PCA. The feed-forward neural network is proposed as classifier. The default neural network structure has been applied, fully connected, with one hidden layer.

In order to demonstrate the effectiveness of the classification, a set of train and test features was prepared according to the described feature extraction method. Based on these, the learning and testing process of the ANN was carried out. The results of the simulation tests carried out are presented in the next section.

## 3. Results

During the study, it was assumed that the signals are observed in fixed frequency range and that the carrier frequency for each analysed signal is estimated. Simulation parameters and their values are presented in Table 1.

Simulation studies included verification of the signal feature extraction process by observing the mutual distribution of the first three PCA components of each waveform type. The distribution of the PCA components is shown in Figure 14 and Figure 15 for SNR taking values from 0 to 10 dB and from −5 to 0 dB, respectively.

Comparing the distribution of the PCA components on Figure 15 and Figure 16, it is clear that for SNR≤0 dB, the features of the individual signals mix more than for signals with SNR≥0 dB.

At first, the features of the LFM and SFM signals overlapped the most. For this reason, during the research, it was decided to use an additional feature. It is related to the CWT frequency profile, and determined on the basis of the number of local maxima.

Figure 16 shows the confusion matrix for the proposed algorithm when considering SNR between 0 and 10 dB. A 100% classification correctness was obtained for all signals except PCCW, where 2% was classified as RCW. Test classification accuracy was equal to 0.9975.

Figure 17 shows the confusion matrix for the proposed algorithm when considering SNR between −5 and 0 dB. There is a noticeable saucer of classification accuracy and so for RCW 93% was correctly classified and 7% was recognised as PCCW, for PCCW 96% was classified correctly and 4% was identified as RCW. Further, for SFMPW 99% was correctly recognised 1% was classified as SFCW. For RPW and PCPW 98% was correctly recognised and 2% was respectively identifies as PCPW and RPW. Test classification accuracy was equal to 0.98.

The last confusion matrix on Figure 18 shows how the classification accuracy of individual signals develops without the additional feature related to the number of maxima of the CWT frequency profile. A slight decrease in classification accuracy for frequency modulated signals, in comparison to confusion matrix presented in Figure 16, is apparent. Test classification accuracy was equal to 0.99.

## 4. Discussion

The simulation results indicate a high classification accuracy (above 99%) of the selected radar signals in particular for SNR above 0 dB. The decrease in performance for SNR < 0 dB particularly relates to the ability to separate the features of CW signals without modulation from those with phase coding. The problem with SFM and LFM signals separation was solved by the use of additional methods, such as an analysis of the occurrence of local extrema of the CWT frequency profile. A key parameter influencing the ability to determine local extrema is the appropriate choice of wavelet type and its parameters to achieve the highest possible resolution in frequency. In presented work Morse wavelet with γ=27 and β=27 is considered.

Furthermore, the method of determining the local extrema of the CWT frequency profile can be used to determine the optimum scales (frequencies) to perform additional CWT frequency cuts and to determine other signal parameters such as the number of frequency levels used in multi-carrier signals. Furthermore, performing CWT frequency cuts at the appropriate scale (frequency) can also improve recognition performance for phase-coded signals. In view of the above, a further development of the algorithm is planned and further simulation tests are to be carried out. At the same time, it is planned to collect real data (signals) in order to confirm the effectiveness of the proposed algorithm in real conditions.

The type of wavelet used also requires further research. The spuriousness of phase-coded signal recognition may require the use of other types of wavelets, e.g., Bump or Paul (analytic Morse with γ=1) which have good time resolution properties.

Analysing the distribution of the PCA components shown in Figure 14 and Figure 15 indicates very good separation using the proposed algorithms between CW and PW signals and within these groups between waveforms with and without frequency modulation.

It is therefore worth considering the application of the proposed algorithm under conditions where SNR > 0 while in the case of large noise interference, its application is limited to a more general classification of signals, e.g., into four groups:CW with FM;CW without FM;PW with FM;PW without FM.

Further analysis is also required to verify the applicability of the algorithm for a wider range of signal parameters such as PRF, PW or code type in PC waveforms.

The undeniable advantage of the proposed algorithm is the ability to use once calculated CWT coefficients for the needs of other algorithms (Figure 19), e.g., to determine the parameters of signals such as PRF, PW, TOA [4], as well as by algorithms using CNN [7,13,14,15], where the CWT coefficients can act as TFI to recognise intra-pulse modulation. The performance of the signal recognition algorithm using CWT, HOS and ANN with the signals used in communication (such as ASK, PSK, FSK and QAM) was confirmed in [17]. Therefore, once computed, the CWT coefficient matrix can be used in parallel by other processing methods, which can significantly affect the functionality of the entire recognition system.

During the simulation tests, an artificial neural network was used with the default network structure, fully connected, with one hidden layer. An open issue that requires further research is the appropriate selection of the ANN structure type and learning model parameters ensuring the highest classification efficiency.

## 5. Conclusions

In this paper recognition algorithm using CWT, HOS and ANN was considered. The obtained results confirm the effectiveness of using algorithms with CWT and ANN when SNR is above 0 dB. The percentage of correct signal type classification is above 99% (test accuracy). The decrease in performance for SNR < 0 dB particularly relates to the ability to separate the features of CW signals without modulation from those with phase coding.

There were the PCA used for feature dimension reduction. The time of signal observation included about 20 pulses. The feed-forward network was considered for decision process.

The method of feature extraction based of CWT and HOS seems to be very versatile. The results presented in [17] shows that this is effective way to recognise signals as ASK, PSK, FSK, QAM and the results presented in this article show, that it is also effective for radar signal classification.

The main advantages of the proposed method include:universality (can be used both in the SIGINT and ELINT area);flexibility related to the possibility of selecting the type and parameters of the wavelet depending on the needs;immediate decision (related to the use of ANN);good separation between pulsed and continuous signals;possibility of parallel use of the calculated matrix of CWT coefficients by other algorithms (e.g., signal/emitter parameter estimation).

In turn, the disadvantages of the proposed solution include:visible decline in classification correctness for SNR close to or lower than 0 dB;the need for using additional methods (CWT frequency profile or histogram) to improve the separation abilities within group of signals with frequency modulation;additional research is required on the influence of the applied wavelet on the possibility of classifying individual types of signals (group of signals with FM, group of signals with phase coding, etc.).

Future work will include wider range of selected signals with specific parameters such as pulse width, pulse envelope, pulse repetition time, intra-pulse modulation, phase coding types, etc. Additional analyses related to the selection of the wavelet type and parameters should be carried out. It is also planned to conduct research with the use of real radar signals.

## Figures and Tables

**Figure 1 sensors-22-07434-f001:**
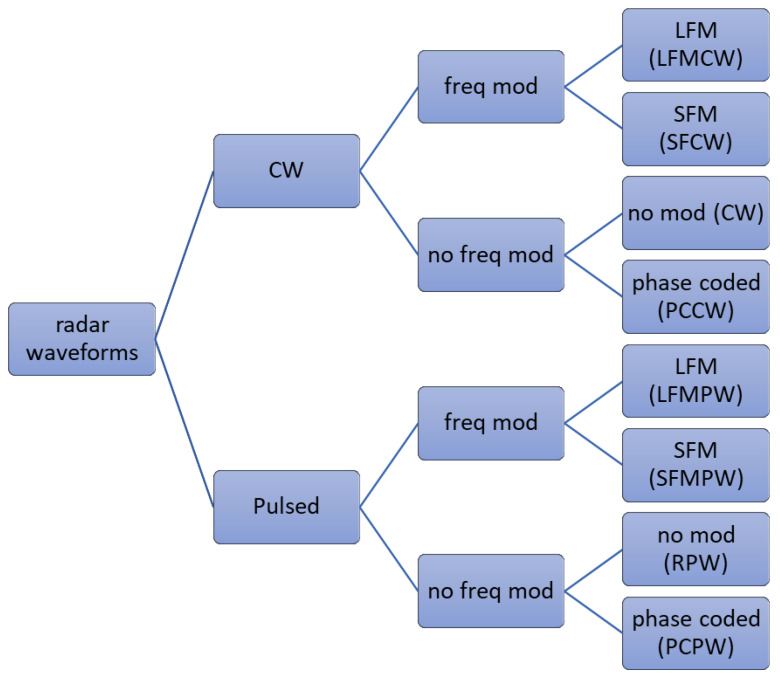
Considered type of waveforms.

**Figure 2 sensors-22-07434-f002:**

Scalogram for the LFMCW signal made using wavelets: (**a**) Morse γ=27, β=27. (**b**) Bump (**c**) Morse (default parameters) γ=3, β=20.

**Figure 3 sensors-22-07434-f003:**

Scalogram for the PCCW signal made using wavelets: (**a**) Morse γ=27, β=27. (**b**) Bump (**c**) Morse (default parameters) γ=3, β=20.

**Figure 4 sensors-22-07434-f004:**

Scalogram for the SFM pulsed signal made using wavelets: (**a**) Morse γ=27, β=27. (**b**) Bump (**c**) Morse (default parameters) γ=3, β=20.

**Figure 5 sensors-22-07434-f005:**

Scalogram for the one pulse phase coded (PC) with Barker code signal made using wavelets: (**a**) Morse γ=27, β=27. (**b**) Bump (**c**) Morse (default parameters) γ=3, β=20.

**Figure 6 sensors-22-07434-f006:**
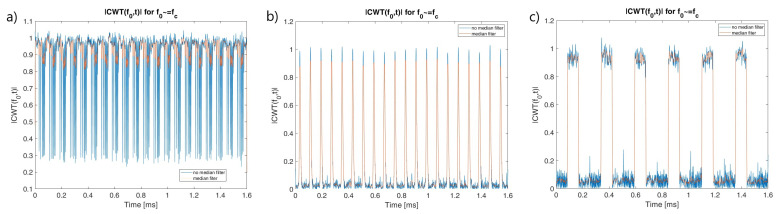
$\left|CWT(f_{0},t)\right|$ for $f_{o}≅f_{c}$ (Morse wavelet used with γ=27, β=27): (**a**) PCCW. (**b**) LFMCW. (**c**) SFMCW.

**Figure 7 sensors-22-07434-f007:**
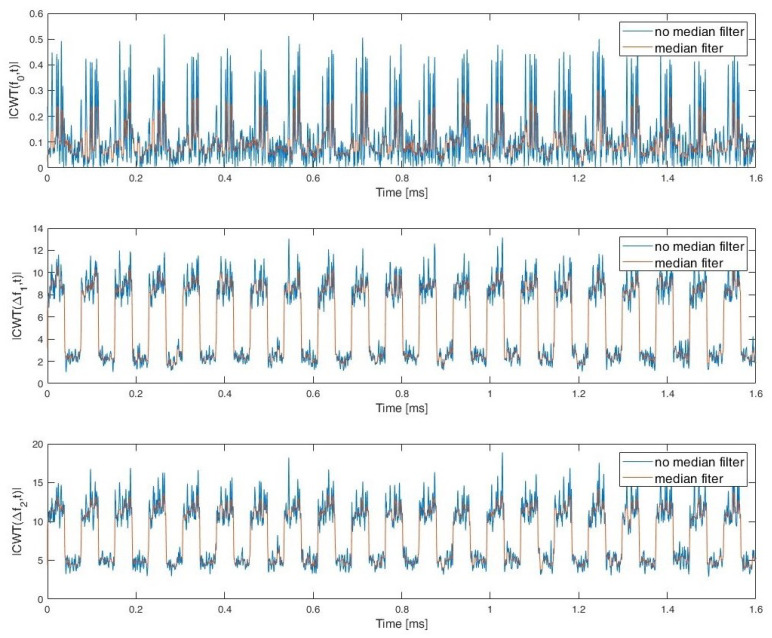
CWT frequency cut for $f_{o}≅f_{c}$ and sum of CWT coefficients range $\Delta f_{1}$ and $\Delta f_{2}$. PCPW.

**Figure 8 sensors-22-07434-f008:**
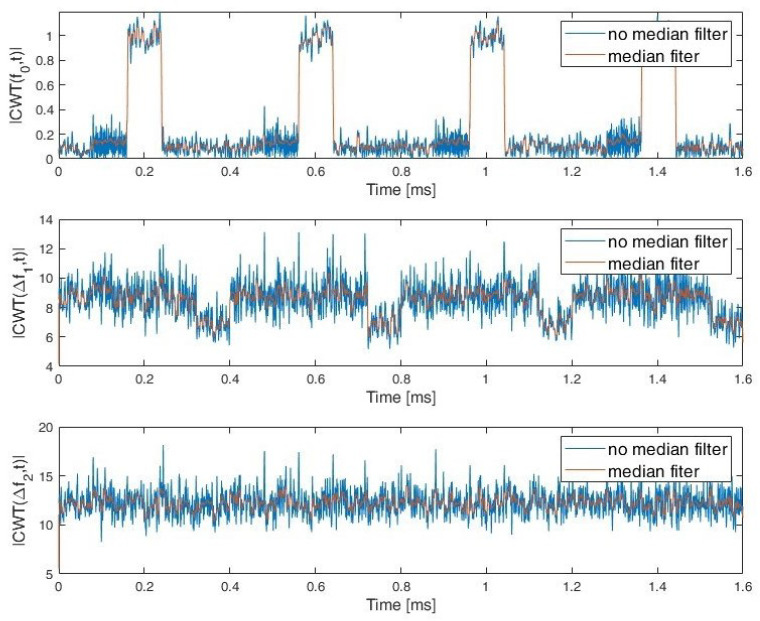
CWT frequency cut for $f_{o}≅f_{c}$ and sum of CWT coefficients range $\Delta f_{1}$ and $\Delta f_{2}$. SFCW.

**Figure 9 sensors-22-07434-f009:**
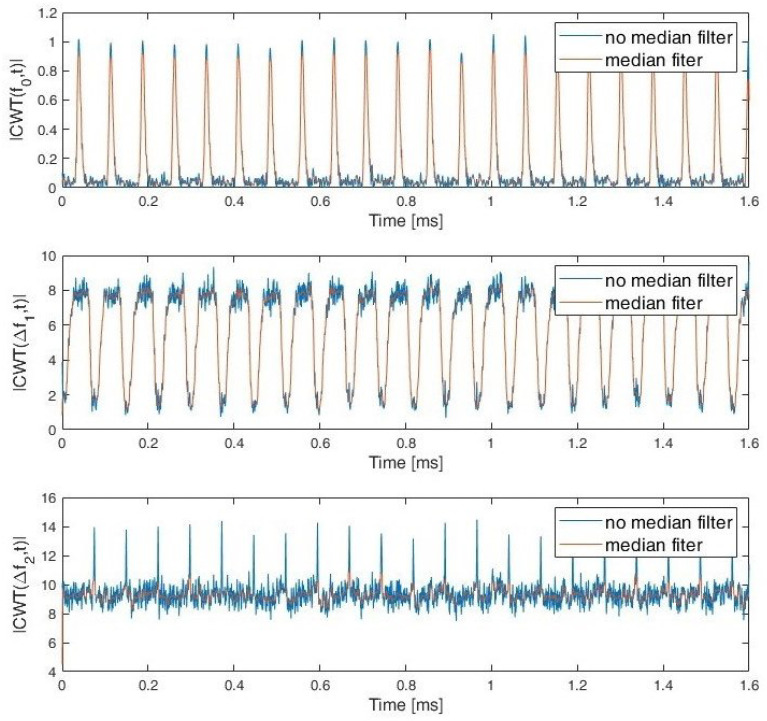
CWT frequency cut for $f_{o}≅f_{c}$ and sum of CWT coefficients in range $\Delta f_{1}$ and $\Delta f_{2}$. LFMCW.

**Figure 10 sensors-22-07434-f010:**
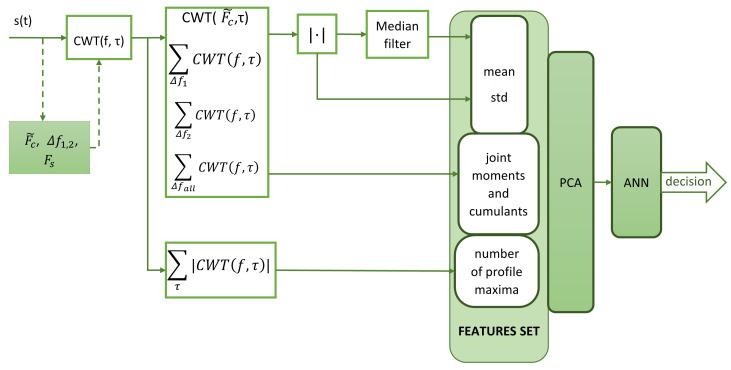
The overall structure of radar signal recognition process.

**Figure 11 sensors-22-07434-f011:**
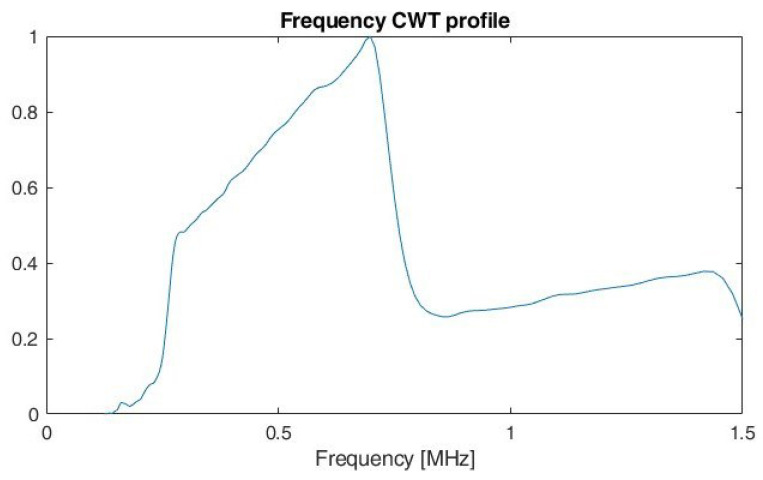
Frequency CWT profile for LFMCW.

**Figure 12 sensors-22-07434-f012:**
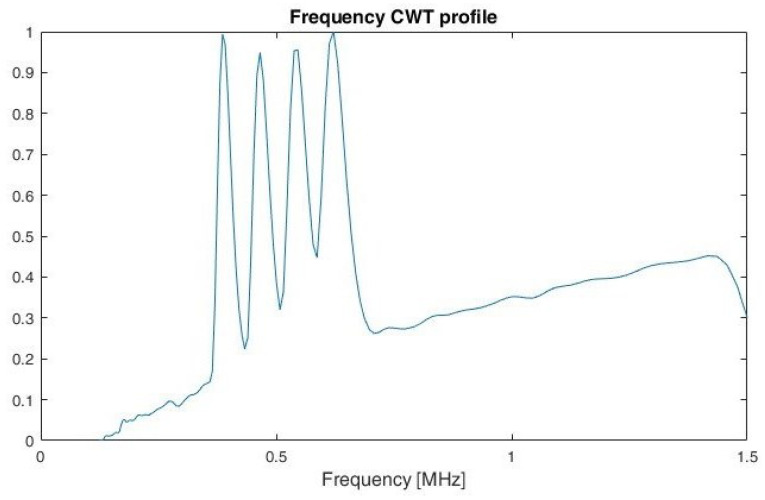
Frequency CWT profile for SFCW.

**Figure 13 sensors-22-07434-f013:**
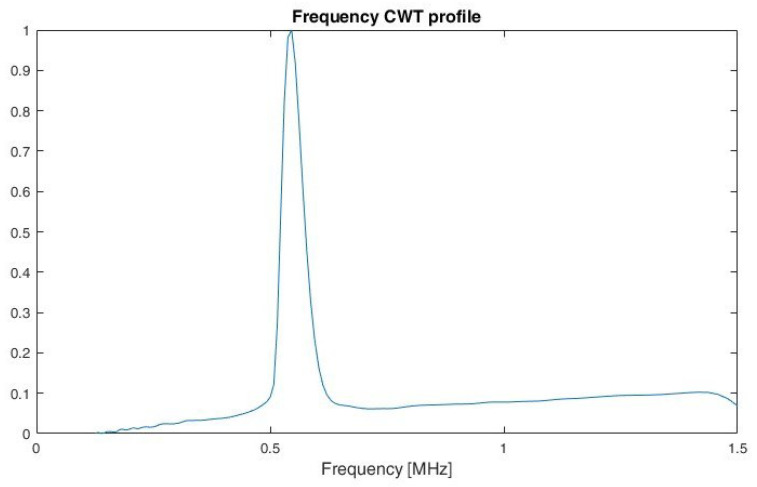
Frequency CWT profile for PCCW.

**Figure 14 sensors-22-07434-f014:**
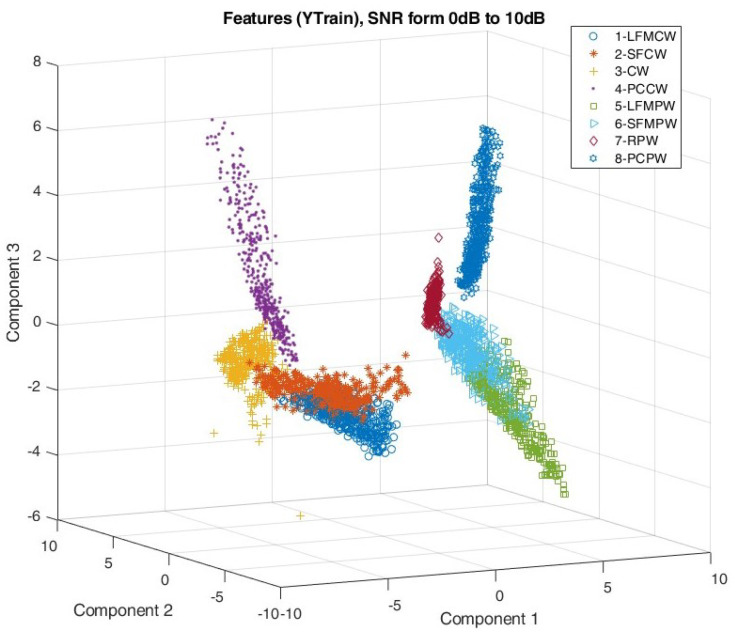
First three components after feature dimension reduction using PCA for SNR in range [0:10] dB.

**Figure 15 sensors-22-07434-f015:**
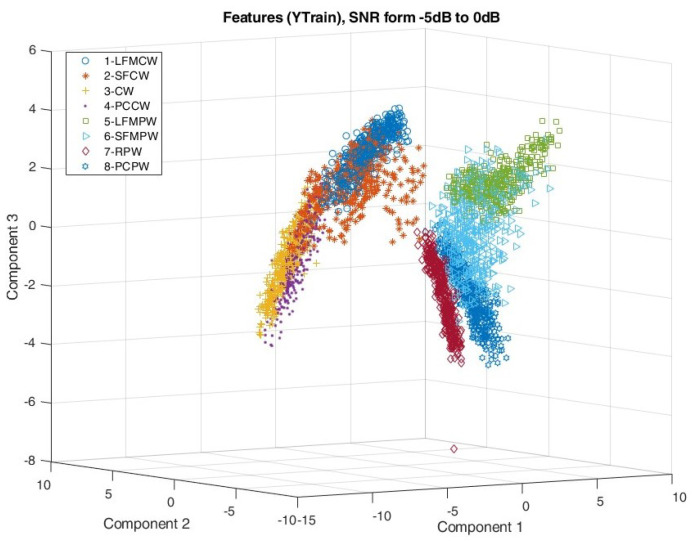
First three components after feature dimension reduction using PCA for SNR in range [−5:0] dB.

**Figure 16 sensors-22-07434-f016:**
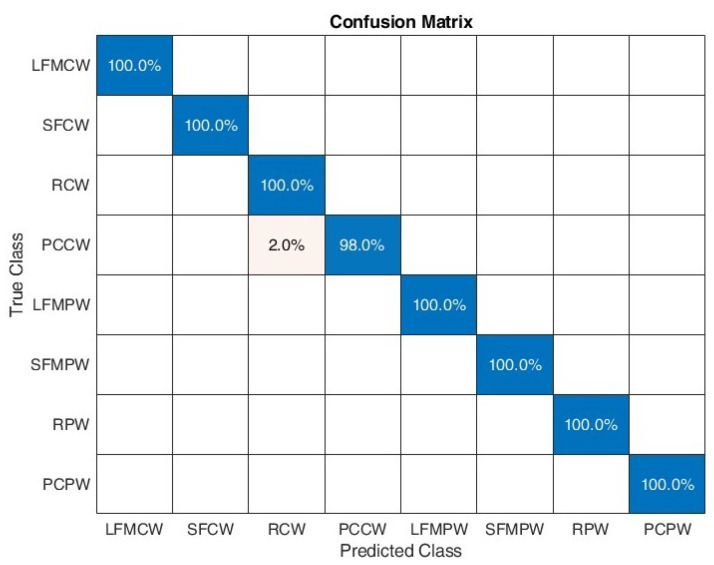
Confusion matrix for SNR in range [0:10] dB.

**Figure 17 sensors-22-07434-f017:**
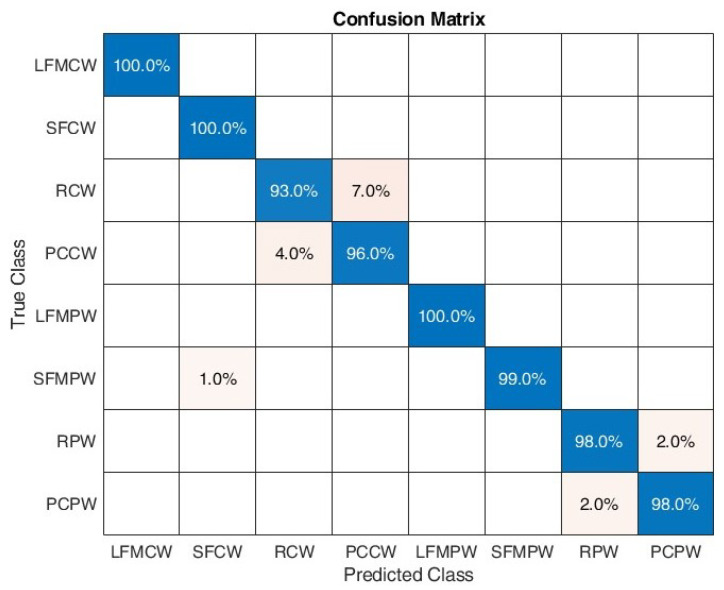
Confusion matrix for SNR in range [−5:0] dB.

**Figure 18 sensors-22-07434-f018:**
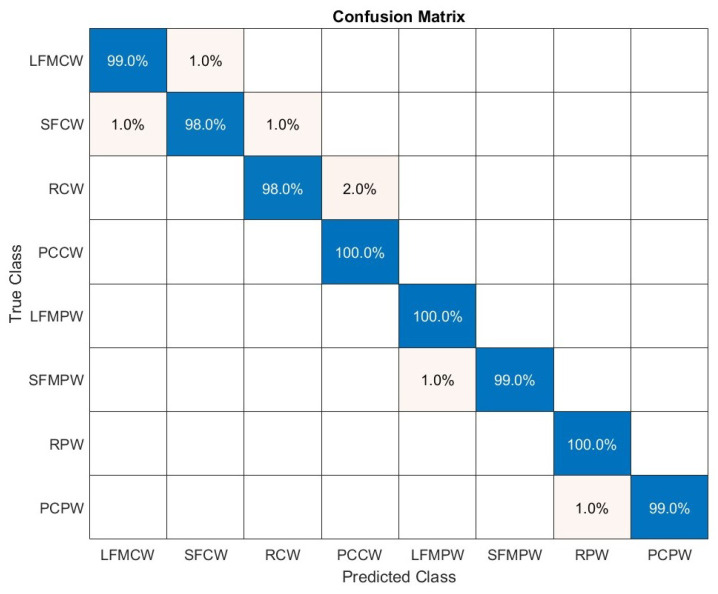
Confusion matrix for SNR in range [0:10] dB for algorithm version without CWT frequency profile (without local maxima detection).

**Figure 19 sensors-22-07434-f019:**
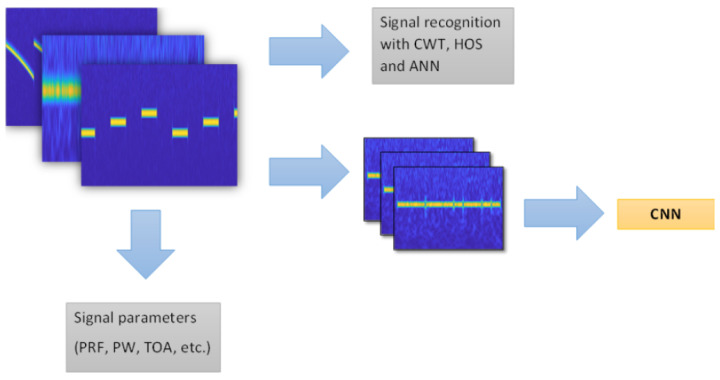
Proposal to use CWT coefficients with other methods used in signal recognition.

**Table 1 sensors-22-07434-t001:** This is a table caption. Tables should be placed in the main text near to the first time they are cited.

Parameter Name	Parameter Value
$F_{c}$	5 MHz ± 0.5 MHz
$F_{s}$	30 MHz
Bandwidth (for LFM Waveforms)	from 2.5 to 5 MHz
Sweep frequency (for SFM Waveforms)	from 0.5 to 1 MHz
SNR	from −5 to 0 [dB] or from 0 to 10 [dB]
Number of pulses in signal	≈20
Pulse Width	≈0.1 ms for pulsed waveforms
Pulse repetition frequency	≈500 Hz
Number of each type of signal	500 (400 for training, 100 for testing)

## Data Availability

Not applicable.

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
