# Peer review of "Application of Continuous Wavelet Transform and Artificial Naural Network for Automatic Radar Signal Recognition"

_sensors, 2022, doi:10.3390/s22197434_

Round 1

Reviewer 1 Report

The Automatic Radar Signal Recognition proposed by the authors is a topic of interest for researchers in related fields, but the paper needs very significant improvements before it can be accepted for publication.

1.  For the innovation point of this work, I found that the author did not describe it in the rticle. As a well-structured paper, the authors should have given the innovation of this paper at the end of the introduction or the related work. The authors' work on certain problems. Or, the improvements made by the authors for certain models. Please summarize a few innovations in the manuscript so that the reader can read it with an accurate understanding of the purpose of this work. Rather than reading the entire paper to get a sense of your true purpose.

2.  It is recommended that in the manuscript, the author should add a vivid simulation or a real picture (or a visio diagram) to describe the intention of the work done by the author and the innovation points, etc.

3.  On page 14 of the manuscript, there is one point that I do not understand. Please ask the authors to check the experimental data carefully. Is there a mistake in the confusion matrix horizontal and vertical coordinates of Figure 18 and Figure 17. Why does Figure 18 add up to 1 vertically while Figure 17 adds up to 1 horizontally?

4.  In the (CONCLUSION) section the authors should elaborate on the shortcomings of the algorithm in this paper. With all due respect, the conclusion should be rewritten to 1) clearly describe the necessary features/advantages of the proposed method that other methods do not have, and 2) describe the limitations of the proposed method and what aspects of the proposed method can be further improved, why and how.

5.  There are several other issues regarding this work as follows.

a)   First, it makes sense for the authors to use a range of methods to classify radar signals. However, why only the classification of various radar simulated signals is covered in the manuscript and no real data is put to the experiment? It is well known that there is a large deviation between the real physical world data and the simulated data. It seems common sense to use the simulated data for classification and identification. Since the authors' goal is to propose a good classification algorithm, they should experiment with real data instead of using simulated data alone. In some specific radar application studies, they usually classify by collecting real radar data, which better reflects the generalization of the model [1][2].

b)  Secondly, when we use radar to collect data indoors, the received signal may contain noise generated by multipath effects, which can be read in the literature [3][4]. These noises affect the spectrogram, so I am rather doubtful that the model proposed by the authors can have the possibility of practical application.

[1] Belgiovane D ,  Chen C C . Micro-Doppler characteristics of pedestrians and bicycles for automotive radar sensors at 77 GHz[C]// 2017 11th European Conference on Antennas and Propagation (EUCAP). IEEE, 2017.

[2] Vandersmissen, B. ,  Knudde, N. ,  Jalalvand, A. ,  Couckuyt, I. ,  Bourdoux, A. , &  Neve, W. D. , et al. (2018). Indoor person identification using a low-power fmcw radar. IEEE Transactions on Geoscience and Remote Sensing, 3941-3952.

[3] Leigsnering M, Ahmad F, Amin M G, et al. Compressive sensing-based multipath exploitation for stationary and moving indoor target localization[J]. IEEE Journal of Selected Topics in Signal Processing, 2015, 9(8): 1469-1483.

[4] Setlur P, Amin M, Ahmad F. Multipath model and exploitation in through-the-wall and urban radar sensing[J]. IEEE Transactions on Geoscience and Remote Sensing, 2011, 49(10): 4021-4034.

Author Response

Dear Reviewer,

Thank you for the article review. The replies to the comments are attached in file Rev1.docx.

Best Regards,

Marta Walenczykowska

Reviewer 2 Report

This paper describes the concept of an algorithm for automatic recognition of selected radar signals: linear frequency modulated pulsed waveform (LFMPW), stepped frequency modulated pulsed waveform (SFMPW), phase coded pulsed waveform (PCPW), rectangular pulsed waveforms (RPW), frequency modulated continuous wave (FMCW), continuous wave (CW), Stepped Frequency Continuous Wave SFCW) and Phase Coded Continuous Waveform (PCCW). The algorithm presented in this paper is based on the use of continuous wavelet transform (CWT) coefficients and higher-order statistics (HOS) in the feature determination of selected signals.

It is a well-structured paper with interesting results. However, it requires further improvements.

(1)   The abstract should be improved. Your point is your own work that should be further highlighted.

(2)   The parameters in expressions are given and explained.

(3)   The main contributions of this paper should be further summarized and clearly demonstrated. This reviewer suggests the authors exactly mention what is new compared with existing approaches and why the proposed approach is needed to be used instead of the existing methods.

(4)   The values of parameters could be a complicated problem itself, how the authors give the values of parameters in the used methods.

(5)   There have generalized Morse, Bump and Morlet. Why you select Bump wavelet?  Please give some reason.

(6)   The literature review is poor in this paper. You must review all significant similar works that have been done. I hope that the authors can add some new references in order to improve the reviews. For example, https://doi.org/10.3390/agriculture12060793; https://doi.org/10.1016/j.isatra.2021.07.017  and so on

Author Response

Dear Reviewer,

Thank you for the article review. The replies to the comments are attached in file Rev2.docx

Best Regards,

Marta Walenczykowska

Round 2

Reviewer 1 Report

Good jobI agree to publish this manuscript. However, authors should carefully check the data and grammar of the article before publication

Reviewer 2 Report

ok